# Maximizing Induced Cardinality Under a Determinantal Point Process

**Jennifer Gillenwater**
Google Research NYC
jengi@google.com

**Alex Kulesza**
Google Research NYC
kulesza@google.com

**Zelda Mariet**
Massachusetts Institute of Technology
zelda@csail.mit.edu

**Sergei Vassilvitskii**
Google Research NYC
sergeiv@google.com

## Abstract

Determinantal point processes (DPPs) are well-suited to recommender systems where the goal is to generate collections of diverse, high-quality items. In the existing literature this is usually formulated as finding the mode of the DPP (the so-called MAP set). However, the MAP objective inherently assumes that the DPP models "optimal" recommendation sets, and yet obtaining such a DPP is nontrivial when there is no ready source of example optimal sets. In this paper we advocate an alternative framework for applying DPPs to recommender systems. Our approach assumes that the DPP simply models user engagements with recommended items, which is more consistent with how DPPs for recommender systems are typically trained. With this assumption, we are able to formulate a metric that measures the expected number of items that a user will engage with. We formalize this optimization of this metric as the Maximum Induced Cardinality (MIC) problem. Although the MIC objective is not submodular, we show that it can be approximated by a submodular function, and that empirically it is well-optimized by a greedy algorithm.

## 1 Introduction

Diversity is frequently advantageous for recommender systems. It can compensate for uncertainty, for example, when a search engine can't be sure which type of "java" a user intended and hence returns results spanning coffee, programming languages, and Indonesia. But diversity can also be an inherently desirable property, reflecting the way that users engage with a set of results. A news feed, for example, might include stories on politics, health, sports, and arts—even when the important news of the day is all political—simply because users enjoy reading a variety of articles. This is one of the reasons why diversity has been a longstanding focus for research on information retrieval and recommender systems [Smyth and McClave, 2001, Herlocker et al., 2004, Ziegler et al., 2005, Hurley and Zhang, 2011].

The determinantal point process (DPP), a probabilistic model of subset selection that prefers diverse sets, is a natural fit for these kinds of applications. However, while DPPs offer efficient algorithms for probabilistic operations like marginalization, conditioning, and sampling, in practice we often need to select a single "best" set, and this can be more challenging. To date, most research in this direction has focused on approximation algorithms for finding the set with the highest probability, sometimes called the *maximum a posteriori* (MAP) set [Gillenwater et al., 2012, Kathuria and Deshpande, 2017, Zhang and Ou, 2016, Nikolov and Singh, 2016]. In particular, the MAP objective has recently

been applied to recommender systems with some success [Chen et al., 2017, Wilhelm et al., 2018]. However, we argue that, for most recommender systems, the MAP objective is not the best fit.

As an alternative, in this paper we propose and analyze induced cardinality (IC), which directly measures the expected number of items that the user will engage with. This is a natural objective for any recommender system where engagements (e.g., clicks) are an important metric. We investigate basic properties of the IC objective for DPPs, finding that it is fractionally subadditive but not submodular, and that, as with MAP, it is NP-hard to find the Maximum Induced Cardinality (MIC) set. Despite this negative result, we are able to establish a data-dependent bound showing that the IC objective can often be well-approximated by a submodular function, which offers corresponding greedy optimization guarantees for the MIC problem. We also show that, empirically, a greedy algorithm typically performs very well.

In the remainder of this section we cover background material on DPPs and discuss why MIC is likely a better fit for recommendation systems than MAP. We proceed in Section 2 to study basic properties of the IC objective, and then in Section 3 we consider the implications of those properties for the MIC optimization problem. Finally, in Section 4 we present empirical studies of several optimization algorithms.

## 1.1 Background

Given an $n \times n$ positive semi-definite (PSD) kernel matrix $L$, the associated determinantal point process assigns to any subset $S$ of $[n] = \{1, 2, \ldots, n\}$ the probability $\mathcal{P}_L(S) = \frac{\det(L_S)}{\det(L+I)}$, where $L_S$ denotes the restriction of $L$ to the row and column indices found in $S$. Note that, because $\sum_{S \subseteq [n]} \det(L_S) = \det(L + I)$, the DPP defined above is a properly normalized probability distribution over all $2^n$ subsets of items drawn from the ground set $[n]$.

**Intuition.** If we think of the diagonal kernel entry $L_{ii}$ as a measurement of the quality of item $i$, then it is not difficult to see that $\mathcal{P}_L$ assigns higher probabilities to sets with high-quality items. If we think of the off-diagonal entry $L_{ij}$ as a scaled measurement of the similarity between items $i$ and $j$, then properties of determinants can be used to show that $\mathcal{P}_L$ assigns higher probabilities to sets whose items are less similar—i.e., more diverse. Thus a DPP prefers sets of items that are both high-quality and diverse. For more background on DPPs, see Kulesza and Taskar [2012].

Given a training collection consisting of subsets of $[n]$, the general DPP learning problem is to find a kernel $L$ such that $\mathcal{P}_L$ best replicates the empirical distribution of the subsets in the training collection. For instance, this can be done using maximum likelihood estimation (MLE), for which a variety of optimization techniques have been developed [Kulesza and Taskar, 2011, Gillenwater et al., 2014, Mariet and Sra, 2015, Dupuy and Bach, 2018, Gartrell et al., 2017]. In this work, we do not attempt to improve upon these learning techniques. Rather, we assume that one of these techniques has been applied to learn a DPP kernel $L$ for a recommendation task, and we focus on how to best use that kernel.

## 1.2 Recommender System Example

Suppose we want to recommend $k$ items from a large set denoted by $[n]$, where $n \gg k$. As training data, we have $r$ examples of previously recommended $k$-sets $[S_1, S_2, \ldots, S_r]$ and the associated user engagements with those sets $[E_1, E_2, \ldots, E_r]$. (That is, each $E_i \subseteq S_i$ is the set of items that a user actually clicked on, watched, read, etc.)

To learn a DPP for this problem, assume we have a parameterized kernel $L(\theta)$, where $\theta$ is the parameter to be learned. (Concretely, you might imagine that each item $i$ has an associated feature vector $\mathbf{b}_i \in \mathbb{R}^d$, and we define the kernel as something like $L_{ij}(\theta) = \mathbf{b}_i^\top \operatorname{diag}(\theta) \mathbf{b}_j$, where $\theta \in \mathbb{R}^d$ and $\operatorname{diag}(\theta)$ denotes the $d \times d$ matrix with $\theta$ on the diagonal. In reality, of course, the kernel may also depend on context such as a query string or the user's history.)

Let $L^{(i)}$ be a shorthand for $L_{S_i}$, the $|S_i| \times |S_i|$ kernel matrix over the set $S_i$. A natural learning problem is to find the value of $\theta$ that maximizes the log-likelihood of the interaction sets $\{E_i\}$ given the recommendation sets $\{S_i\}$:

$$\max_\theta \sum_{i=1}^r \log \left( \mathcal{P}_{L^{(i)}(\theta)}(E_i) \right) = \max_\theta \sum_{i=1}^r \log \det \left( L^{(i)}(\theta)_{E_i} \right) - \log \det \left( L^{(i)}(\theta) + I \right) . \quad (1)$$

Optimization techniques from the references in the previous section can be applied to learn a good $\theta$ under this learning objective.

At test time, we want to generate new recommendations using a DPP with kernel parameterized by this fixed $\theta$. Previous work has addressed the problem of generating a recommendation set of size $k$ by (approximately) solving the following optimization problem:

**Problem 1** (Maximum *a posteriori* (MAP))**.**

$$\max_{S:|S|=k} \mathcal{P}_{L(\theta)}(S) = \max_{S:|S|=k} \det(L_S(\theta)) \tag{2}$$

At first glance, this MAP objective seems quite natural: it seeks the most likely set under the probabilities defined by the DPP. However, it has subtle semantics. Recall that at training time we learned $\theta$ by maximizing the likelihood of the items that were *engaged* with, not the likelihood of all items that were recommended. That is, we maximized the probability of $\{E_i\}$, not $\{S_i\}$. (It wouldn't make any sense to maximize the likelihood of $\{S_i\}$, as this would result in learning a DPP that simply mimics whatever recommender system was used in generating $\{S_i\}$ in the first place.) Thus, the learned DPP models user engagements, not recommended sets. Formally, this means that the MAP objective $\mathcal{P}_L(S) = \frac{\det(L_S)}{\det(L+I)}$ represents the probability that a user, when presented with a recommendation consisting of *all* of the available items, will engage with *every* item in $S$.

In practice, of course, it is the set $S$ that gets shown to the user, who then engages with some subset of $S$. Hence, the MAP objective does not have the correct semantics, instead introducing a mismatch between train and test time.

## 1.3 Maximum Induced Cardinality

As an alternative to MAP, we propose Maximum Induced Cardinality (MIC), which appeals to a specific notion of success that is natural for many recommender systems: maximizing the number of recommended items that the user engages with. Whereas with the MAP objective the ground set is $[n]$ and the modeled variable is the set $S \subseteq [n]$ of recommended items, here the ground set is $S$ and the modeled variable is the set $E \subseteq S$ of recommended items that the user eventually *engages* with:

$$\mathcal{P}_{L_S}(E) = \frac{\det(L_E)}{\det(L_S + I)} . \tag{3}$$

This matches the learning setup above. The MIC problem, which aims to maximize the expected cardinality of the induced engagement set $E$, can be formalized as follows.

**Problem 2** (Maximum Induced Cardinality (MIC))**.**

$$\max_{S:|S|=k} f(S), \quad f(S) \equiv E_{E \sim \mathcal{P}_{L_S}}[|E|] . \tag{4}$$

While $f(S)$ is naïvely an exponential sum over all $2^k$ subsets of $S$, it can be simplified:

$$f(S) = \sum_{E \subseteq S} |E| \mathcal{P}_{L_S}(E) = \sum_{E \subseteq S} |E| \frac{\det(L_E)}{\det(L_S + I)} = \text{Tr}(I - (L_S + I)^{-1}) . \tag{5}$$

The final equality follows from Equations 15 and 34 in Kulesza and Taskar [2012]. In this form, the time required to compute $f(S)$ is dominated by the inverse, which requires $O(k^3)$ time.

## 1.4 MAP vs MIC

As discussed above, the semantics of MIC are more appropriate for recommender systems than those of MAP. However, the choice of objective can also have immediate, practical consequences. While both MIC and MAP can produce relatively diverse sets in general, when the DPP kernel is low-rank MAP can fail dramatically. This occurs in practice, for instance, if there are a small number of features relative to the size of the desired recommendation set.

To illustrate how MAP fails and why MIC does not, let's consider a toy example (see Figure 1). Suppose that we are in a movie recommendation context, and each dot in Figure 1

represents one movie, with the size of the dot proportional to movie quality. Suppose further that the two dimensions in Figure 1 are star ratings and box office revenue. Then the three clusters correspond to three types of movies: 1. "Artistic gems" characterized by high ratings but low revenue; 2. "Oscar winners" with high ratings and high revenue; and 3. "Summer blockbusters", which have low critical ratings but high revenue. Each of these categories could be desirable, depending on the user's mood, so it might be advantageous to recommend one movie from each group. However, when asked to select a set of size $k = 3$, the MAP objective has equal value (zero) for all size-three sets —with only two features, the rank of the DPP kernel is 2, and hence the determinant of any $3 \times 3$ matrix will be zero. Hence MAP cannot distinguish among any three-item sets. The MIC objective on the other hand continues to provide useful differentiation even when the number of items requested exceeds the rank of the kernel matrix. It will select one item from each cluster (e.g., the + items), whereas MAP will select a random size-3 set (e.g., the x items).

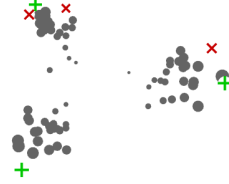

Figure 1: Example where MIC (+) is more diverse than MAP (x).

## 2  Properties of Induced Cardinality

We begin by presenting key properties of the IC objective. We show that while $f(S)$ is monotone (Theorem 1), it is not submodular (Example 1.1), but is fractionally subadditive (Theorem 2). These results will inform the subsequent discussion of optimization techniques in Section 3.

**Theorem 1.** $f(S)$ *is monotone increasing.*

Showing that $f(S)$ is monotone is a straightforward application of the Cauchy eigenvalue interlacing theorem. The proof can be found in the supplement.

We can also show that $f(S)$ is not submodular. Formally, recall that a set function $g$ is submodular if for all sets $S \subseteq T \subseteq [n]$ and all $i \notin T$ :

$$g(S \cup \{i\}) - g(S) \geq g(T \cup \{i\}) - g(T) . \tag{6}$$

To show non-submodularity, it suffices to create a single counterexample violating this property.

**Example 1.1.** *Consider $n = 3$ items, and define a matrix $F$ with one row of features per item:*

$$F = \begin{bmatrix} 2 & 0 \\ 2 & 0 \\ \sqrt{2} & \sqrt{2} \end{bmatrix}, \quad L = FF^\top = \begin{bmatrix} 4 & 4 & 2\sqrt{2} \\ 4 & 4 & 2\sqrt{2} \\ 2\sqrt{2} & 2\sqrt{2} & 4 \end{bmatrix}$$

*Let $S = \{1\}$, $T = \{1, 2\}$, and $i = 3$; then it is easy to verify that the inequality required for submodularity, Equation 6, does not hold.*

To give some intuition, recall the original definition of $f(Z)$ as the expected set size under $\mathcal{P}_{L_Z}$. When $Z = S = \{1\}$, $\mathcal{P}_{L_Z}$ is split between the empty set and the singleton $\{1\}$. When $Z = T = \{1, 2\}$, the probability is still only split between the empty set and singletons, because items 1 and 2 are identical and so $\det(L_T) = 0$. Hence, $f(T)$ is not much larger than $f(S)$. However, when $Z = T \cup \{i\}$, both $\{1, 3\}$ and $\{2, 3\}$ have substantial probability mass, whereas $S \cup \{i\}$ only supports a single size-2 subset, $\{1, 3\}$. Hence $f(T \cup \{i\})$ ends up being substantially larger than $f(S \cup \{i\})$.

It is also possible to construct examples showing that Equation 6 does not hold even approximately.

**Example 1.2.** *Define the feature matrix*

$$F = \begin{bmatrix} x & x \\ x & x + 1 \\ 1 & 1 \end{bmatrix}$$

*for a given value $x$, and let $L = FF^\top$ as before. Then, for $S = \{1\}$, $T = \{1, 2\}$, and $i = 3$, one can verify that $\frac{f(T \cup \{i\}) - f(T)}{f(S \cup \{i\}) - f(S)}$ grows without bound as $x \to \infty$.*

We note that there does exist a restricted setting of $L$ where $f(S)$ is provably submodular. Recall that a real matrix $L$ is an M-matrix if all of its off-diagonal entries are non-positive, and all of its eigenvalues are non-negative. Theorem 3 of Friedland and Gaubert [2013] shows that $f(S)$ is submodular whenever the kernel matrix $L$ is an M-matrix. In practice, for many applications, the off-diagonal entries of the kernel matrix are naturally positive, and in these cases, the kernel is not an M-matrix. In Section 4 we consider algorithms that first project to the kernel to an M-matrix and then optimize it using the standard greedy submodular maximization algorithm.

While $f(S)$ is not submodular, it is (fractionally) subadditive. Recall that a set function $g$ on $[n]$ is:

- **Subadditive** if for all sets $S, T \subseteq [n]$: $g(S \cup T) \leq g(S) + g(T)$ .
- **Fractionally subadditive** if $g(S) \leq \sum_i \alpha_i g(T_i)$ for all $T_i \subseteq [n]$ and all constants $0 \leq \alpha_i \leq 1$ such that $\sum_{i:j \in T_i} \alpha_i \geq 1$ for all $j \in S$. Note that a fractionally subadditive function is also subadditive.

**Theorem 2.** $f(S)$ *is fractionally subadditive.*

The proof can be found in the supplement.

## 3   Optimizing Induced Cardinality

In the previous section we showed that $f(S)$ is monotone and subadditive, but not submodular. In contrast to monotone submodular functions, for which the greedy algorithm [Nemhauser et al., 1978] is guaranteed to give a $(1 - 1/e)$-approximation, subadditive functions cannot be approximated by general black box methods [Feige, 2009]. Moreover, we have the following result:

**Theorem 3.** MIC *is NP-hard.*

The proof can be found in the supplement. Despite its NP-hardness, however, we will develop an approximation algorithm for MIC. We begin by giving a different representation of the objective function, expressing it as an infinite geometric series. We show that the first few terms of the series are submodular, and can thus be optimized using greedy methods. By bounding the contribution of the remaining terms we can then prove a data-dependent approximation bound.

**Geometric Series Representation.**   Recall from Equation 5 that $f(S) = |S| - \operatorname{Tr}((L_S + I)^{-1})$. Denote the largest eigenvalue of $L$ by $\lambda_n(L)$. Then, define the PSD matrix $B = (m-1)I - L$, with $m = \lambda_n(L) + 1$. (The smallest eigenvalue of $B$ will be zero, and the largest will be at most $m - 1$.) Re-arranging, we have: $L + I = mI - B$. Since $\lambda_n(B/m) < 1$, we can apply the Neumann series representation [Suhubi, 2003, page 390] to this expression:

$$m(L + I)^{-1} = \left( I - \frac{1}{m}B \right)^{-1} = \sum_{i=0}^{\infty} \frac{B^i}{m^i} \ . \tag{7}$$

Thus, we can re-write $f(S)$ as an infinite sum of traces of matrix powers:

$$f(S) = |S| - \sum_{i=0}^{\infty} \frac{\operatorname{Tr}(B_S^i)}{m^{i+1}} \ . \tag{8}$$

Note that $B_S^i$ here means $(B_S)^i$ and not $(B^i)_S$.

**Submodularity.**   The first two terms in this sum are modular functions:

$$\frac{\operatorname{Tr}(B_S^0)}{m} = \frac{|S|}{m} \quad \text{and} \quad \frac{\operatorname{Tr}(B_S)}{m^2} = \sum_{i \in S} \frac{B_{ii}}{m^2} \ . \tag{9}$$

Corollary 2 in Friedland and Gaubert [2013] states that the third term, $\frac{\operatorname{Tr}(B_S^2)}{m^3}$, is a supermodular function. Thus, the following function, consisting of the first few terms from the geometric series representation of $f$, is submodular:

$$\hat{f}(S) = |S| - \frac{|S|}{m} - \frac{\operatorname{Tr}(B_S)}{m^2} - \frac{\operatorname{Tr}(B_S^2)}{m^3} \ . \tag{10}$$

**Monotonicity.** This function is also monotone. This is easiest to see by expressing it in terms of $f$. Let $h$ represent the difference between $\hat{f}$ and $f$:

$$h(S) = \sum_{i=3}^{\infty} \frac{\mathrm{Tr}(B_S^i)}{m^{i+1}} \; . \tag{11}$$

Then $\hat{f}(S) = f(S) + h(S)$. Since $f$ is monotone, it remains to show that $h$ is monotone. Consider sets $S, T$ such that $S \subseteq T$. Then, by the Cauchy eigenvalue interlacing theorem, the $j$-th eigenvalue of $B_S$ is smaller than the $(j + |T| - |S|)$-th eigenvalue of $B_T$. Hence, $\mathrm{Tr}(B_S) \leq \mathrm{Tr}(B_T)$, and similarly for all higher powers of these matrices. Thus, $h$ is monotone and so is $\hat{f}$.

We propose to maximize $\hat{f}$ using the standard greedy algorithm [Nemhauser et al., 1978], which we will refer to as GREEDY. Since $\hat{f}$ is monotone submodular, this gives a $(1 - 1/e)$ approximation; that is, let $\hat{S}$ be the solution returned by GREEDY, and let $\hat{S}^*$ be the true maximizer of $\hat{f}$. Then:

$$\hat{f}(\hat{S}) \geq (1 - 1/e)\hat{f}(\hat{S}^*) \; . \tag{12}$$

**Tail Analysis.** To show that $\hat{S}$ is a good approximation for MIC, it remains to bound the difference between $\hat{f}$ and $f$. Recall that $h$ represents this difference. Let $B_S = QAQ^{-1}$ be the eigendecomposition of the PSD matrix $B_S$. Note that $B_S B_S = QAQ^{-1}QAQ^{-1} = QA^2Q^{-1}$, and hence $B_S^i = QA^iQ^{-1}$ is also a PSD matrix. This means that $h$ is non-negative. Thus, $f(S) = \hat{f}(S) - h(S) \leq \hat{f}(S)$.

We will show (Theorem 4) that $\hat{f}$ is also bounded by $f$ from above in that there exists some constant $0 < c \leq 1$ such that $c\hat{f}(S) \leq f(S)$. Combining these inequalities with Equation 12:

$$f(\hat{S}) \geq c\hat{f}(\hat{S}) \geq c(1 - 1/e)\hat{f}(\hat{S}^*) \geq c(1 - 1/e)\hat{f}(S^*) \geq c(1 - 1/e)f(S^*) \; .$$

Thus, the final approximation ratio achieved by this procedure is $c(1 - 1/e)$. It remains to prove a bound on $c$. We start by proving a theorem that bounds $c$ for a particular set $S$, later extending it to a uniform result as a corollary.

**Theorem 4.** *The ratio of $f$ (Equation 8) to $\hat{f}$ (Equation 10) is bounded from below:*

$$\frac{f(S)}{\hat{f}(S)} \geq 1 - \frac{mr(B_S, 3)}{(m-1)k - r(B_S, 1) - r(B_S, 2)} \; , \quad \text{where } r(B_S, \ell) = \sum_{j=1}^{|S|} \left( \frac{\lambda_j(B_S)}{m} \right)^{\ell} , \tag{13}$$

$k = |S|$, $m = \lambda_n(L) + 1$, *and* $B = (m-1)I - L$.

*Proof.* The ratio of interest is $\frac{f(S)}{\hat{f}(S)} = 1 - \frac{h(S)}{\hat{f}(S)}$. We lower-bound it by substituting an upper bound for $h$.

$$h(S) = \sum_{i=3}^{\infty} \frac{\mathrm{Tr}(B_S^i)}{m^{i+1}} = \sum_{i=3}^{\infty} \frac{\sum_{j=1}^{k} \lambda_j(B_S)^i}{m^{i+1}} = \frac{1}{m} \sum_{j=1}^{k} \sum_{i=3}^{\infty} \left( \frac{\lambda_j(B_S)}{m} \right)^i \tag{14}$$

$$= \frac{1}{m} \sum_{j=1}^{k} \left( \frac{\lambda_j(B_S)}{m} \right)^3 \left( \frac{1}{1 - \frac{\lambda_j(B_S)}{m}} \right) \leq r(B_S, 3) \; , \tag{15}$$

$$\tag{16}$$

where the last equality follows from the geometric series summation formula, and the last inequality follows by definition of $r$ and the fact that $\lambda_j(B_S) \leq m - 1$.

Noting that:

$$\hat{f}(S) = k - \frac{k}{m} - \frac{\mathrm{Tr}(B_S)}{m^2} - \frac{\mathrm{Tr}(B_S^2)}{m^3} = \left( 1 - \frac{1}{m} \right) k - \frac{1}{m}[r(B_S, 1) + r(B_S, 2)] \; , \tag{17}$$

we can now substitute the upper bound on $h$ to complete the proof. $\square$

**Corollary 4.1.** *For all sets $S$ of size $k$,*

$$\frac{f(S)}{\hat{f}(S)} \geq 1 - \frac{mr'(B,k,3)}{(m-1)k - r'(B,k,1) - r'(B,k,2)} \, , \quad \text{with } r'(B,k,\ell) = \sum_{j=n-k+1}^{n} \left(\frac{\lambda_j(B)}{m}\right)^{\ell} . \tag{18}$$

The proof of the corollary can be found in the supplement. The value of $c$ given by Corollary 4.1 is best (closest to 1) for matrices $B$ where the eigenvalues are small, which will be the case when eigenvalues of $L$ are close to $\lambda_n(L)$. In the extreme case where $L$ is approximately a multiple of the identity matrix, $B$ and thus $r'(B,k,\ell)$ will be close to zero. In this case, $c \approx 1$.

The value of $c$ is worst when the eigenvalues of $B$ decay slowly, which means that most eigenvalues of $L$ are small compared to $\lambda_n(L)$. In the extreme case where all of the top-$k$ eigenvalues of $B$ are identical and equal to $m - 1$, the expression for $c$ is :

$$c = 1 - \frac{(m-1)^2}{m^2 - m - 1} = \frac{m}{(m-1)^2 + m} \approx \frac{1}{m} \, , \tag{19}$$

and thus the approximation is less meaningful. However, in contrast to the MAP objective, this degradation of approximation is gradual, and catastrophic failure such as that seen in Figure 1 is completely avoided.

## 4 Experiments

As described in Section 1, MIC's semantics are a better fit for DPP-based recommendation systems, whereas the traditional application of MAP leads to a mismatch between how the DPP is learned and how it is applied. Properly comparing MIC to MAP on real data requires a live system where we can observe users engaging with different sets of recommendations; a static dataset is not likely to be sufficient since the number of possible recommendation sets is combinatorially large. (See the work by Swaminathan et al. [2017] for a longer discussion of the challenges here.) In this work, we focus on evaluating algorithms that optimize the MIC objective, specifically evaluating the GREEDY algorithm in three settings: 1) when optimizing $f(S)$, 2) when optimizing $f(S)$ after projecting to the space of M-matrices (see Section 4.1 for details), and 3) when optimizing the submodular approximation $\hat{f}(S)$. We call these methods and their results GIC, PIC, and SIC respectively.

### 4.1 Projecting to the set of $M$-matrices

We tried several methods for projecting to the set of (real, symmetric) PSD M-matrices for the PIC method. We found that flipping the signs of any positive off-diagonal elements, then projecting to the PSD cone by truncating negative eigenvalues at zero worked best. If the PSD projection resulted in any positive off-diagonal elements, we simply iterated the process of flipping their signs and projecting to the PSD cone until the resulting matrix satisfied all requirements.

Note that the sign-flipping step computes a projection onto the set of Z-matrices (under the Frobenius norm). Since the set of Z-matrices is closed and convex, as is the set of PSD matrices, this means that the iterative process described above is guaranteed to converge. (Though it will not necessarily converge to the projection onto the intersection of the two convex sets.)

### 4.2 Runtime Analysis

GIC: The definition of $f(S)$ in Equation 5 implies that in iteration $i$ of GREEDY we need to compute an $i \times i$ matrix inverse to evaluate the objective on each of the remaining items. Rather than doing this directly, requiring time $O(nk^4)$, we can use incremental inverse updates Hager [1989]. This reduces the runtime of GREEDY by a factor of $k$ to $O(nk^3)$. (Note that PIC's runtime is identical, ignoring the initial step of projecting to the space of M-matrices.)

SIC: At first glance, $\hat{f}(S)$ requires squaring an $i \times i$ matrix for each item (Equation 10). This too can be substantially improved by taking advantage of the fact that $\text{Tr}(B_S B_S) = \sum_{s_1 \in S} \sum_{s_2 \in S} b_{s_1 s_2}^2$. Evaluating a prospective point simply requires updating this sum with $i$ new terms. This reduces the naïve runtime of GREEDY by a factor of $k^2$ to $O(nk^2)$, making SIC a factor of $k$ faster than GIC.

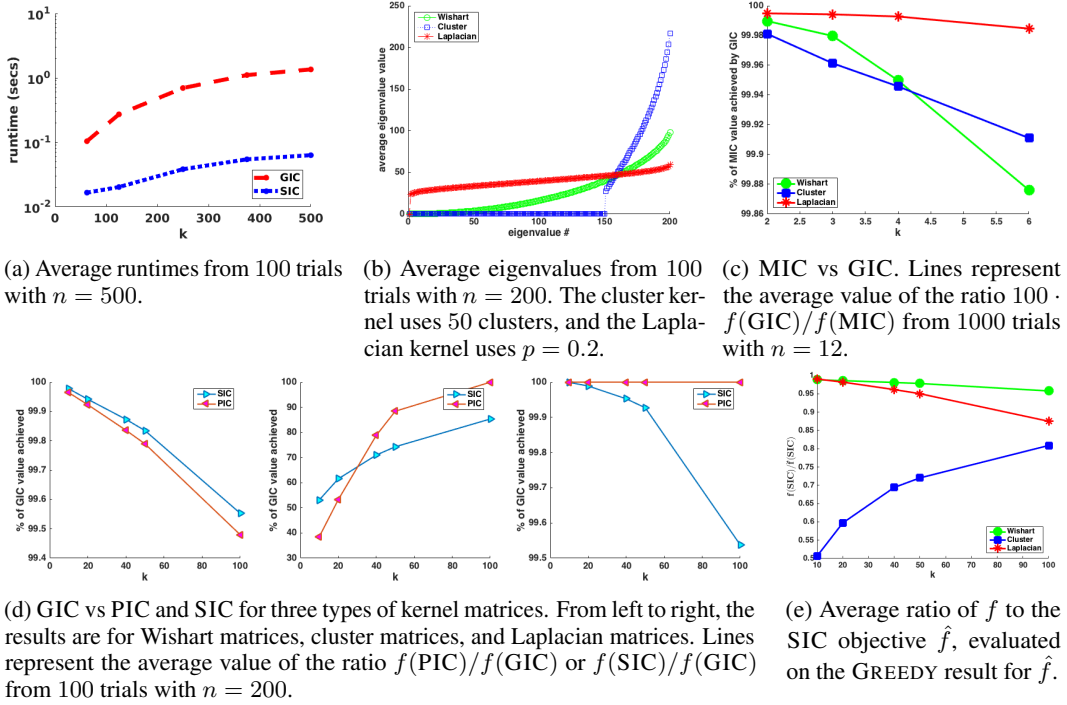

(a) Average runtimes from 100 trials with $n = 500$.

(b) Average eigenvalues from 100 trials with $n = 200$. The cluster kernel uses 50 clusters, and the Laplacian kernel uses $p = 0.2$.

(c) MIC vs GIC. Lines represent the average value of the ratio $100 \cdot f(\text{GIC})/f(\text{MIC})$ from 1000 trials with $n = 12$.

(d) GIC vs PIC and SIC for three types of kernel matrices. From left to right, the results are for Wishart matrices, cluster matrices, and Laplacian matrices. Lines represent the average value of the ratio $f(\text{PIC})/f(\text{GIC})$ or $f(\text{SIC})/f(\text{GIC})$ from 100 trials with $n = 200$.

(e) Average ratio of $f$ to the SIC objective $\hat{f}$, evaluated on the GREEDY result for $\hat{f}$.

Figure 2: Experimental results

Figure 2a shows the runtimes for GIC and SIC. For $n = 500$ and $k = 250$, SIC runs about 18 times faster.

## 4.3 Approximation Quality

We ran experiments with three types of kernel matrices:

- Wishart matrix: For each item, draw a feature vector from an $n$-dimensional zero-mean Gaussian. Stack the feature vectors into a matrix $F$, and set $L$ to $FF^\top$.
- Cluster matrix: Divide items evenly into $k$ clusters, and sample an $n$-dimensional mean from $\mathcal{N}(0, 1)$ for each cluster. Draw each item's feature vector from $\mathcal{N}(\mu, 1)$, where $\mu$ is the corresponding cluster mean. Stack the feature vectors into a matrix $F$, and set $L$ to $FF^\top$.
- Graph Laplacian: Generate an $n$-node random graph using the Erdos-Renyi random graph model with edge existence probability $p$. Compute the graph Laplacian matrix from the degree matrix, $D$, and the adjacency matrix, $A$: $L = D - A$.

As Figure 2b shows, each of these three types of matrices has a distinct shape to its spectrum. The Wishart grows rapidly but smoothly. The cluster matrix also has rapid, smooth growth for $k$ of its eigenvalues (one per cluster), but has value zero for all other eigenvalues. The Laplacian has a smooth, nearly linear growth, the slope of which generally increases with $p$.

### 4.3.1 Comparison to MIC

Although GIC does not, in general, have any approximation guarantees, empirically we found that it was quite effective. In Figure 2c we plot the ratio of the GREEDY solution, GIC, to the optimum, MIC (for small $n$ where it is possible to compute MIC by brute force). GIC does best on the Laplacian matrices (edge existence parameter is fixed at $p = 0.2$), and slightly worse on the other two matrix types. The success of GIC on the Laplacian may be partly due to the fact that Laplacians are M-matrices, and, as mentioned in Section 2, $f(S)$ is submodular in this case. The performance on the Wishart and cluster kernels is not quite as good, but GIC still achieves more than 99% of the maximum possible value in both cases.

Note that it is only possible to compute MIC for relatively small $n$, since it requires an exhaustive search of the space of all $\binom{n}{k}$ possible size-$k$ subsets. Thus, in all subsequent experiments we use the GIC solution as the baseline for comparing with PIC and SIC.

### 4.3.2 Comparison to PIC and SIC

In general, the M-matrix projection, PIC, and the submodular approximation, SIC, slightly underperform GIC. This is despite a formal approximation guarantee on the performance of SIC. Figure 2d shows the performance of the methods on each of the three types of kernels.

- For Wishart matrices, SIC does slightly better than PIC, and both methods are consistently finding good sets whose value is at least 99% that of the GIC.
- For cluster matrices, SIC and PIC struggle, sometimes choosing sets with less than half the value of GIC.
- For Laplacian matrices, PIC is identical to GIC. This is because Laplacian matrices are M-matrices, and hence $L$ does not need to be projected. The SIC results are also consistently good. In the plot, we show values for Laplacians with edge existence parameter $p = 0.01$ rather than the $p = 0.2$ used in earlier experiments, as for $p = 0.2$ the spectrum is non-flat enough that SIC results are indistinguishable from PIC and GIC.

The SIC trends in Figure 2d can be explained by the extent to which $f$ is well-approximated by the SIC objective, $\hat{f}$. In Figure 2e we plot the ratio of the two. Note that for Wishart and Laplacian matrices, the $f/\hat{f}$ ratio decays slowly with $k$, hence optimizing $\hat{f}$ is very similar to optimizing $f$. For the cluster matrices, the ratio grows dramatically with $k$, which explains the poor performance of SIC for low values of $k$.

## 5 Conclusion

Our proposed MIC optimization problem has advantages over the common MAP setup for recommender systems in terms of interpretability and train-test time matching. In this work we have shown that the MIC objective can often be well-approximated by a submodular function and optimized by a straightforward greedy algorithm. Future work includes the application of MIC to real-world recommendation systems.

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
