[Supplementary Material]

# Supplemental Material:
# Maximizing Induced Cardinality Under a
# Determinantal Point Process

**Jennifer Gillenwater**
Google Research NYC
jengi@google.com

**Alex Kulesza**
Google Research NYC
kulesza@google.com

**Zelda Mariet**
Massachusetts Institute of Technology
zelda@csail.mit.edu

**Sergei Vassilvitskii**
Google Research NYC
sergeiv@google.com

## 1   Monotonicity of Induced Cardinality

The monotonicity of $f$ is a straightforward application of the Cauchy eigenvalue interlacing theorem.

**Theorem 1.** *Cauchy interlacing theorem: Consider a symmetric matrix $A \in \mathbb{R}^{n \times n}$ with eigenvalues $\alpha_1 \leq \alpha_2 \leq \ldots \leq \alpha_n$, and any principal submatrix $B \in \mathbb{R}^{m \times m}$ with eigenvalues $\beta_1 \leq \beta_2 \leq \ldots \leq \beta_m$. Then the eigenvalues interlace in the following manner:*

$$\alpha_k \leq \beta_k \leq \alpha_{k+n-m} \qquad \text{for } k = 1, 2, \ldots, m \, . \tag{1}$$

**Theorem 2.** $f(S)$ *is monotone increasing.*

*Proof.* Given two sets $S \subseteq T \subseteq [n]$, we will show that $f(S) \leq f(T)$. Denote the eigenvalues of $L_T$ by $\alpha_1 \leq \alpha_2 \leq \ldots \leq \alpha_{|T|}$, and the eigenvalues of $L_S$ by $\beta_1 \leq \beta_2 \leq \ldots \leq \beta_{|S|}$. Since $L_S$ is a principal submatrix of $L_T$, the Cauchy interlacing theorem implies that:

$$\alpha_k \leq \beta_k \leq \alpha_{k+|T|-|S|} \qquad \text{for } k = 1, 2, \ldots, |S| \, . \tag{2}$$

We will combine this fact with the eigenvalue version of the formula for $f(S)$ to get the desired inequality:

$$
\begin{aligned}
f(S) &= |S| - \sum_{i=1}^{|S|} \frac{1}{\beta_i + 1} \leq |S| - \sum_{i=1}^{|S|} \frac{1}{\alpha_{i+|T|-|S|} + 1} \\
&\leq |S| - \sum_{i=1}^{|S|} \frac{1}{\alpha_{i+|T|-|S|} + 1} + \sum_{i=1}^{|T|-|S|} \left( 1 - \frac{1}{\alpha_i + 1} \right) \\
&= |T| - \sum_{i=1}^{|S|} \frac{1}{\alpha_{i+|T|-|S|} + 1} - \sum_{i=1}^{|T|-|S|} \frac{1}{\alpha_i + 1} \\
&= |T| - \sum_{i=1}^{|T|} \frac{1}{\alpha_i + 1} = f(T) \, ,
\end{aligned}
$$

where the first inequality is an application of the interlacing theorem, and the second follows because the quantity being added is positive ($0 \leq \alpha_i$ since $L$ is a positive semi-definite matrix).  $\square$

## 2 Fractional Subadditivity of Induced Cardinality

Before proving fractional subadditivity, we state two matrix algebra facts that will be useful in completing the proof. The first follows from the definition of the adjugate matrix, and the second from the submodularity of $\log \det$.

**Lemma 2.1.** *For an invertible matrix $A \in \mathbb{R}^{n \times n}$, $\mathrm{Tr}(A^{-1}) = \sum_{i=1}^{n} \det(A_{-i})/\det(A)$, where $A_{-i} \in \mathbb{R}^{(n-1) \times (n-1)}$ is the $A$ matrix with its ith row and column removed.*

*Proof.* The adjugate matrix $\mathrm{adj}(A)$ is defined as the matrix that satisfies $\det(A)I = A\mathrm{adj}(A)$. This matrix is known to be the transpose of the cofactor matrix of $A$. More concretely, let $M_{ij}$ denote the $(i,j)$-minor of the matrix $A \in \mathbb{R}^{n \times n}$ (the determinant of the matrix formed by deleting row $i$ and column $j$ from $A$). Let $C_{ij} = (-1)^{i+j} M_{ij}$ denote the corresponding cofactor. The adjugate matrix $\mathrm{adj}(A)$ equals $C^\top$. Plugging $C^\top$ into the definition of the adjugate: $\det(A)I = AC^\top$. Multiplying by $A^{-1}$ and taking the trace:

$$\det(A)\,\mathrm{Tr}(A^{-1}) = \sum_{i=1}^{n}(-1)^{i+i} M_{ii} = \sum_{i=1}^{n} \det(A_{-i}). \tag{3}$$

Dividing by $\det(A)$ yields the desired identity. $\qquad \square$

Second, from the submodularity of $\log \det$, we have the following useful lemma.

**Lemma 2.2.** *For any set $T_i \subseteq T$ and any $j \in T_i$:*

$$\frac{\det(L_{T_i \setminus j} + I)}{\det(L_{T_i} + I)} \leq \frac{\det(L_{T \setminus j} + I)}{\det(L_T + I)}. \tag{4}$$

*Proof.* The function $g(S) = \log \det(L_S + I)$ is a well-known submodular function. From the definition of submodularity, we can say that for any set $T_i \subseteq T$ and any $j \in T_i$, the following inequality holds:

$$g(T_i) - g(T_i \setminus j) \geq g(T) - g(T \setminus j). \tag{5}$$

Writing out the expression for $g$ and combining logs:

$$\log\left(\frac{\det(L_{T_i} + I)}{\det(L_{T_i \setminus j} + I)}\right) \geq \log\left(\frac{\det(L_T + I)}{\det(L_{T \setminus j} + I)}\right).$$

Exponentiating and then taking the inverse yields the desired expression. $\qquad \square$

We are now ready to prove fractional subadditivity of $f(S)$.

**Theorem 3.** *$f(S)$ is fractionally subadditive.*

*Proof.* Let $S$, $T_i$, and $\alpha_i$ satisfy the relationship $\sum_{i:j \in T_i} \alpha_i \geq 1$ for all $i \in S$. Then $\{T_i\}$ must cover $S$, in the sense that $S \subseteq \bigcup_i T_i$. (If some element $s \in S$ were not in any of the $T_i$, then we would have $\sum_{i:s \in T_i} \alpha_i = 0$.) Hence, by monotonicity (Theorem 2), $f(S) \leq f\left(\bigcup_i T_i\right)$. Let $T = \bigcup_i T_i$ and

$M = L_T$. Then we can write:

$$f(S) \le f(T) = \mathrm{Tr}(I - (M + I)^{-1}) \qquad\qquad \text{(definition of } f\text{)}$$

$$= \sum_{j \in T} \left( 1 - \frac{\det(M_{-j} + I)}{\det(M + I)} \right) \qquad\qquad \text{(by Lemma 2.1)}$$

$$\le \sum_{j \in T} \left( \sum_{i : j \in T_i} \alpha_i \right) \left( 1 - \frac{\det(M_{-j} + I)}{\det(M + I)} \right) \qquad\qquad \text{(multiplication by value} \ge 1\text{)}$$

$$= \sum_i \alpha_i \sum_{j \in T} \left( 1 - \frac{\det(M_{-j} + I)}{\det(M + I)} \right) \qquad\qquad \text{(switching the order of the summations)}$$

$$\le \sum_i \alpha_i \sum_{j \in T} \left( 1 - \frac{\det(L_{T_i \setminus j} + I)}{\det(L_{T_i} + I)} \right) \qquad\qquad \text{(by Lemma 2.2)}$$

$$= \sum_i \alpha_i \sum_{j \in T_i} \left( 1 - \frac{\det(L_{T_i \setminus j} + I)}{\det(L_{T_i} + I)} \right) \qquad\qquad \text{(dropping zero terms)}$$

$$= \sum_i \alpha_i \, \mathrm{Tr}(I - (L_{T_i} + I)^{-1}) \,. \qquad\qquad \text{(by Lemma 2.1)}$$

Thus, $f(S) \le \sum_i \alpha_i f(T_i)$. $\qquad\qquad\square$

# 3 NP-Hardness of Induced Cardinality

Before showing that $f(S)$ is NP-hard to maximize, we first state a lemma that will be helpful in this proof.

**Lemma 3.1.** *For a vector* $\boldsymbol{\lambda} \in \mathbb{R}^k$, *define the function* $h(\boldsymbol{\lambda}) = \sum_{i=1}^k \frac{\lambda_i}{\lambda_i + 1}$. *Then* $\boldsymbol{\lambda} = \mathbf{1}$ *is the unique maximizer of the following optimization problem:*

$$\max_{\boldsymbol{\lambda}} h(\boldsymbol{\lambda}) \ \ s.t. \ \sum_{i=1}^k \lambda_i = k \,. \qquad\qquad (6)$$

*Proof.* Introducing a Lagrange multiplier $\alpha$ for the equality constraint, we have the following Lagrangian function:

$$\mathcal{L}(\boldsymbol{\lambda}, \alpha) = h(\boldsymbol{\lambda}) - \alpha \left( k - \boldsymbol{\lambda}^\top \mathbf{1} \right) \,. \qquad\qquad (7)$$

According to the method of Lagrange multipliers, if $\boldsymbol{\lambda}^*$ is a maximizer of $h$ for the original constrained problem, then there exists an $\alpha^*$ such that $(\boldsymbol{\lambda}^*, \alpha^*)$ is a stationary point of $\mathcal{L}$. Hence, a maximizer of the original problem must occur at a point where all of the partial derivatives of $\mathcal{L}$ are zero. These derivatives are:

$$\frac{\partial \mathcal{L}}{\partial \lambda_i} = \frac{1}{(\lambda_i + 1)^2} - \alpha \quad \text{and} \quad \frac{\partial \mathcal{L}}{\partial \alpha} = k - \sum_{i=1}^k \lambda_i \qquad\qquad (8)$$

For it to be the case that $\frac{\partial \mathcal{L}}{\partial \lambda_i} = 0$ for all $i$, it must be true that $\frac{\partial \mathcal{L}}{\partial \lambda_i} = \frac{\partial \mathcal{L}}{\partial \lambda_j}$ for all $i, j$. This can only hold if $(\lambda_i + 1)^2 = (\lambda_j + 1)^2$, which is only true when $\lambda_i = \lambda_j$. So at any stationary point of $\mathcal{L}$ it must be the case that $\boldsymbol{\lambda}$ is a uniform vector. Satisfying $\frac{\partial \mathcal{L}}{\partial \alpha} = 0$ sets the scale of this vector, requiring $\lambda_i = 1 \ \forall i$. $\qquad\square$

We are now ready to prove the main NP-hardness result.

**Theorem 4.** MIC *is NP-hard.*

*Proof.* Recall the EXACT 3-COVER (X3C) problem: Given a set $S$ and a collection $C$ of size-3 subsets of $S$, decide if there is a sub-collection $C' \subseteq C$ that contains every element of $S$ exactly once.

We will reduce this to MIC with the following construction. (The DPP kernel construction is identical to that of Theorem 2.4 in Kulesza [2012].) First, define a $|C| \times |S|$ matrix $F$ with entries $F_{cs} = \mathbf{1}(s \in C_c)\frac{1}{\sqrt{3}}$. Let $L = FF^\top$. This is a $|C| \times |C|$ positive semi-definite matrix with entries:

$$L_{ij} = \frac{|C_i \cap C_j|}{3} \ .$$

Set $k = \frac{|S|}{3}$. We will now show that MIC $\geq \frac{k}{2}$ if and only if an exact 3-cover exists.

- If an exact 3-cover exists, then MIC $\geq \frac{k}{2}$: Without loss of generality, let $\{C_1, \ldots, C_k\}$ be a collection of size-3 sets that make up an exact 3-cover, and let $Y = \{1, \ldots, k\}$. Then, by construction, for $i, j \in Y, i \neq j$, we have $L_{ij} = 0$ and $L_{ii} = 1$; that is, $L_Y$ is the identity matrix. From the equation for $f$ in terms of eigenvalues, with $\lambda_i$ denoting the eigenvalues of $L_Y$, we have:

$$f(Y) = \sum_{i=1}^{k} \frac{\lambda_i}{\lambda_i + 1} = \sum_{i=1}^{k} \frac{1}{1 + 1} = \frac{k}{2} \ . \tag{9}$$

- If an exact 3-cover does not exist, then MIC $< \frac{k}{2}$: We begin by constructing a relaxation, RELAXED-MIC. The MIC solution can be no larger than the solution to RELAXED-MIC. We will then show that RELAXED-MIC has maximum value $\frac{k}{2}$, and that any solution to MIC that also achieves this value must correspond to an exact 3-cover.

First, note that the diagonal of $L$ is all 1's, so the trace of any size-$k$ principal submatrix is $k$. Since the trace of a matrix is also the sum of its eigenvalues, we have that for any size-$k$ set $Y$:

$$\mathrm{Tr}(L_Y) = \sum_{i=1}^{k} \lambda_i = k \ , \tag{10}$$

where $\lambda_i$ are the eigenvalues of $L_Y$. Now recall the eigenvalue form of $f(S)$:

$$f(Y) = \sum_{i=1}^{k} \frac{\lambda_i}{\lambda_i + 1} \ . \tag{11}$$

If we combine this expression with the constraint on the sum of the eigenvalues, then we get a relaxation of the MIC problem:

RELAXED-MIC :

$$\max_{\lambda_1, \ldots, \lambda_k} \sum_{i=1}^{k} \frac{\lambda_i}{\lambda_i + 1} \qquad \text{s.t.} \sum_{i=1}^{k} \lambda_i = k \ . \tag{12}$$

This problem is identical to MIC, but with fewer constraints, since it does not require that the $\lambda_i$ exactly match the eigenvalues of a size-$k$ submatrix of $L$. From Lemma 3.1, we know that the unique maximizer of this relaxed problem is $\lambda_1 = \ldots = \lambda_k = 1$. This solution has value $\frac{k}{2}$. Thus, the value achieved by MIC must be $\leq \frac{k}{2}$.

We now argue that MIC only achieves this value when there is a submatrix of $L$ that corresponds to an exact 3-cover. Since $\boldsymbol{\lambda} = 1$ is the *unique* maximizer of RELAXED-MIC, no other setting of $\boldsymbol{\lambda}$ can achieve a value as large as $\frac{k}{2}$. Given the construction of the matrix $L$, the only size-$k$ submatrix with exactly this set of all-1 eigenvalues is the identity matrix. But if there exists a size-$k$ submatrix of $L$ that is the identity matrix, then this corresponds to an exact 3-cover, which contradicts the premise that no exact 3-cover exists.

$\square$

## 4 Proof of Uniform Approximation Bound

**Corollary 4.1.** *For all sets $S$ of size $k$,*

$$\frac{f(S)}{\hat{f}(S)} \geq 1 - \frac{mr'(B, k, 3)}{(m-1)k - r'(B, k, 1) - r'(B, k, 2)} \ , \quad \text{with } r'(B, k, \ell) = \sum_{j=n-k+1}^{n} \left(\frac{\lambda_j(B)}{m}\right)^{\ell} \ . \tag{13}$$

*Proof.* By the Cauchy eigenvalue interlacing theorem, the eigenvalues of $B_S$ interlace those of $B$ such that: $\lambda_i(B) \leq \lambda_i(B_S) \leq \lambda_{i+n-k}(B)$. Thus:

$$\text{Tr}(B_S) = \sum_{j=1}^{k} \lambda_j(B_S) \leq \sum_{j=1}^{k} \lambda_{j+n-k}(B) = \sum_{j=n-k+1}^{n} \lambda_j(B) \,. \tag{14}$$

Hence the trace of $B_S$ is upper-bounded by the sum of the top $k$ eigenvalues of $B$. Since raising a matrix to a power simply raises its eigenvalues to that power, we also have:

$$\text{Tr}(B_S^i) \leq \sum_{j=n-k+1}^{n} \lambda_j(B)^i \,, \tag{15}$$

and therefore $r(B_S, \ell) \leq r'(B, k, \ell)$.

We can now substitute $r'(B, k, \ell)$ for $r(B_S, \ell)$ in the original theorem, noting that these terms have positive coefficients in the numerator and negative coefficients in the denominator. $\qquad\square$

## References

A. Kulesza. *Learning with Determinantal Point Processes*. PhD thesis, University of Pennsylvania, 2012.