[Reviews · NeurIPS 2018]

Reviewer 1



The paper argues that Determinantal point process (DPP) simply models user engagements with recommended items instead of the conventional notion where the maximum a posteriori (MAP) set from the DPP is regarded as the collection containing diverse, high quality items and is suggested in recommender systems. The authors advocate an alternative framework for applying DPPs to recommender systems where they an optimize an metric named Maximum Induced Cardinality (MIC) that measures the expected number of items that a user will engage with and assert that this metric is more consistent with how DPPs for recommender systems are typically trained. Some properties of MIC are investigated where it is shown that MIC is not submodular. A close submodular approximation to MIC with theoretical guarantees is instead considered and optimized based by the standard greedy algorithm. The paper is very well written and easy to understand. The arguments made in the paper are very convincing. The approximation to MIC based on Neumann series representation is cogent. Couple of concerns: 1. In line 196 it is claimed that the function \hat{f} is also monotone but no supportive arguments are provided for the same. Submodularity alone does not guarantee the bounds in equation (11). 2. As f(S) is monotone and f(\emptyset) = 0, setting c = 0 in line 203 will satisfy the inequality c \hat{f}(S) <= f(S). So the constant "c" does exist. If the authors in Theorem 4 mean that there exist an \epsilon >0 such that c>=\epsilon, the sentence needs to reworded accordingly.

Reviewer 2



EDIT: I've read the response and I am satisfied with the answers. I maintain my grade. # Summary of the paper and its contributions Determinantal point processes (DPPs) have been investigated in recommendation systems. In that framework and once a DPP has been learned, the authors remark that finding the sample with maximum DPP likelihood (the so-called "MAP recommendation") does not lead to meaningful recommendations. The contributions are as follows. The authors introduce another utility function, the maximum induced cardinality (MIC), and explain how to approximately optimize it using submodular approximations. Algorithms to optimize the MIC criterion are compared on synthetic datasets. # Summary of the review The contribution is interesting and novel, and the paper rather well-written. I have some suggestions to improve clarity in Section 1, namely when describing the data-generating process, see major comments below. My main concern is the lack of real-world datasets that would allow comparing the MIC and the MAP in practice, which would empirically validate the significance of the contribution. However, the semantics of the MIC are rather well-explained, and clearly make more intuitive sense than the MAP, so maybe the lack of experimental validation is OK. # Major comments - Section 1.2: this section can gain in clarity. For instance, it is not clear whether any of the considered sets is actually drawn from DPP(L). From what I gather, there is a big matrix L, and given a set R of recommended items, we model the set S of engaged items as a draw from a DPP with kernel a *submatrix* L_R of L indexed by R. Since you have many independent copies of this procedure (one for each R_i), the loglikelihood becomes (1). But strictly speaking, the DPP with matrix L is never explicitly used. - One way to clarify this is maybe to use [Kulesza and Taskar 2012, Section 2.4.3 "Conditioning"], and model each S as a draw from DPP(L) conditioned to lie within a given subset R. This seems to lead to (1) as well. Then the interest of the maximum induced cardinality later on is clearer: Find the set R such that conditionally on R, DPP(L) has the biggest expected cardinality. - Section 1.2: From what I understand, there is a W matrix of size m by m and you are learning a matrix L = BWB' which is n by n, of rank less than m. This should be maybe underscored in Section 1.2, as you only explicitly define the matrices L_R for the various Rs, but not the parametrization of L. This is minor. - Relatedly to L-ensembles, maximizing (1) will lead to W picking up the cardinality of each set of engaged items. In the end, it is hard to interpret W as it both encodes the importance of each feature in the measure of diversity, and the distribution of the number of engaged items. This is a general intepretational issue with L-ensembles, not with this particular paper. L-ensembles are fine if you consider k-DPPs of mixtures thereof as in [Kulesza and Taskar 2012], but once you want to also model the number of items with the kernel matrix, you run into parametrization trouble. Maybe everything would be easier to interpret and parametrize if you used the correlation matrix K to define the underlying DPP, see also previous bullets. - Section 1.3: should the variable S not be renamed R? What you are trying to model is the set Y in S of engaged items, so I thought the set S is the set of recommended items. If this is the case, I would rename it as R, for consistency with Section 1.2. - Section 4.2: maybe an experiment with a real recommendation set and actual features would be interesting. A minima, why are the three types of matrices you consider good proxies for the L matrix in a recommendation task? # Minor comments & typos - L65: I think [Urschel et al 2017] is advocating the method of moments rather than maximum likelihood. - L79: to each feature

Reviewer 3



The authors have addressed most of my concerns in their response, and I hope to see the changes they mentioned reflected in the final version. ---------- The paper proposes an alternative objective for diverse subset selection using determinant point processes (DPPs). For the task of selecting a best subset from the ground set, the authors argue that instead of the maximum a posterior (MAP) objective, a more suitable objective is to maximize the expected number of items in the subset that the user will engage with, which is termed the maximum induced cardinality (MIC) objective. The paper proceeds to investigate theoretical properties of the MIC objective, and proposes an approximation algorithm for optimizing MIC by observing that the objective, while not sub-modular, is fractionally sub-additive. Overall, the paper is clearly written and appears technically sound. However, while I tend to agree with the intuition that the MIC might be a more suitable objective than MAP in practice, my major concern is that this hypothesis has not been verified/justified with any empirical evidence in the experiments. I believe that this is an important aspect as the hypothesis is not a widely acknowledged fact, and its plausibility greatly affects the significance of the theoretical and methodological contributions made in this paper; thus at least some form of evidence should be provided to justify the motivation of the paper. I understand that as the authors mentioned on line 228, a truly sound comparison between MIC and MAP should be done via a live system on real data, but I believe that simpler simulations or experiments could be enough to shed some light and provide some empirical support for the hypothesis and convince readers of its potential practical importance. Such experiments could be performed on simulated or offline data (e.g., website recommendations/transactions history), or using crowdsourcing tools (e.g., Amazon Mechanical Turk). Some of the evaluation methods used in the DPP papers (Kulesza et al., Gillenwater et al., among others) could also be borrowed here. On the technical side, I have some more specific comments: - The authors noted on line 115 that when the DPP kernel is low-rank, MAP can fail dramatically. However, by Corollary 4.1, it seems that when L is low-rank, some eigenvalues of B could be large, which would result in a small value of c, indicating poor approximation quality of \hat{f} to f? - Line 169: in the summand, what is the index j for? - In terms of running time complexity, how does the approximate MIC algorithms (GIC, PIC, and SIC) compare with the state-of-the-art approximate MAP algorithm? If SIC is significantly more efficient than the current MAP algorithm, this could also be a potential reason to prefer MIC over MAP. - Since PIC was used in the experiments, I feel that at least some details regarding the projection onto the space of PSD M-matrices should be included in the main text. In Section 5 of the supplementary material, the authors states that they found “flipping the signs of any positive off-diagonal elements, then projecting to the PSD cone by truncating negative eigenvalues at zero worked best”, and “if the PSD projection resulted in any positive off-diagonal elements, we simply iterated the process of flipping their signs and projecting to the PSD cone until the resulting matrix satisfied all requirements”. I’m not sure if the sign-flipping is a standard/theoretically justified approach of projecting on to the space of M-matrices, and whether the latter iterative procedure would be guaranteed to terminate? - Minor point: In the experiments using the graph Laplacian, I feel that instead of/in addition to a simple Erdos-Renyi model, using a more realistic random graph model (e.g., the stochastic blockmodel, the preferential-attachment model, etc.) that obeys the many properties found in real-world networks (power-law degree distributions, clustering, small-world property, etc.) could be more informative.